Development and validation of a prognostic nomogram for predicting poor outcomes following intravenous rt-PA in patients with acute ischemic stroke

Zhang Fengjiao 1 2
Zhao Dan 1 2
Zhang Jing xiaoyan20110316@126.com 1
Department of Neurology, Tianjin NanKai Hospital, Tianjin Medical University , Tianjin , China
Department of Neurology, Hospital of Integrated Chinese and Western Medicine , Tianjin , China
Nunes-da-Fonseca Rodrigo
Electronic publication date: 2025 Feb 26
Publication date: 2025
Volume: 13
Electronic Location ID: e18937
Received 2024 Nov 7; Accepted 2025 Jan 14
Copyright: ©2025 Zhang et al.
Copyright year: 2025
Copyright holder: Zhang et al.
License: This is an open access article distributed under the terms of the Creative Commons Attribution License, which permits unrestricted use, distribution, reproduction and adaptation in any medium and for any purpose provided that it is properly attributed. For attribution, the original author(s), title, publication source (PeerJ) and either DOI or URL of the article must be cited.
License URL: https://creativecommons.org/licenses/by/4.0/

Keywords: AIS, Prognostic nomogram, rt-PA, Retrospective Cohort study

Funding: The authors received no funding for this work.

==============================
Background

Intravenous administration of recombinant tissue plasminogen activator (rt-PA) within 4.5 h of symptom onset is a standard treatment for acute ischemic stroke (AIS). However, certain patients continue to develop unfavorable outcomes despite timely rt-PA therapy. Identifying those at high risk is essential for developing individualized care plans and establishing appropriate follow-up.

Methods

This retrospective study included AIS patients treated with intravenous rt-PA at 0.9 mg/kg at our center. Outcomes at three months were evaluated using the modified Rankin Scale (mRS). Patients with mRS scores ≤2 were considered to have favorable outcomes, and those with scores >2 were considered to have poor outcomes. Univariable analysis and stepwise logistic regression were used to identify independent predictors of poor prognosis, and a nomogram was subsequently developed. The model’s discriminative power was assessed with area under the receiver operating characteristic curves (AUC-ROC), and its calibration was examined using calibration plots. Decision curves and clinical impact curves were applied to determine clinical utility.

Results

Among 392 enrolled patients, 77 had poor outcomes three months after rt-PA therapy. Fibrinogen (Fg), baseline NIHSS, and a history of hypertension emerged as independent predictors of poor prognosis. The nomogram achieved an AUC of 0.948 (95% CI [0.910–0.985]), with sensitivity of 0.900 and specificity of 0.916 in the training dataset, and an AUC of 0.959 (95% CI [0.907–1.000]), with sensitivity of 0.943 and specificity of 0.947 in the validation dataset. Calibration plots demonstrated close agreement between predicted and observed probabilities, and decision curves indicated a wide range of net benefit threshold probabilities.

Conclusions

This nomogram, incorporating baseline NIHSS, Fg, and a history of hypertension, accurately predicts poor three-month outcomes in AIS patients treated with intravenous rt-PA. Its ease of use may facilitate early risk stratification and assist clinicians in formulating more targeted management strategies and follow-up protocols for patients likely to experience unfavorable outcomes.

Introduction

Acute ischemic stroke (AIS), which is associated with high incidence, mortality, disability, and recurrence rates, has become the leading cause of disability and death in China (Wang et al., 2017). Intravenous thrombolysis with recombinant tissue plasminogen activator (rt-PA) is currently recommended as a standard therapeutic strategy to restore reperfusion and improve functional outcomes in patients with AIS. Although most patients benefit from this approach, some continue to develop poor outcomes, characterized by persistent neurological deficits and markedly reduced quality of life. These deficits place considerable financial and psychological burdens on both patients and their families (Warner et al., 2019; Emberson et al., 2014; Wardlaw, Sandercock & Berge, 2003). Accurate prediction of a patient’s prognosis after rt-PA therapy is therefore essential for developing personalized intervention strategies and improving clinical outcomes.

Historically, prognostic assessment often involved complex formulas that were impractical for routine clinical use. Today, clinicians mainly rely on the NIHSS score to predict outcomes, but its predictive accuracy remains limited. In recent years, several new prognostic models have been introduced in China and elsewhere; however, many have proven controversial after validation (Nisar, Hanumanthu & Khandelwal, 2019; Li et al., 2015). Moreover, many of these models incorporate numerous variables, rendering them cumbersome in day-to-day practice. To date, there is no widely accepted risk prediction model for identifying factors that lead to poor outcomes after intravenous rt-PA. Consequently, there is a clear clinical need for a simple, rapid, and accurate prediction tool. Nomograms offer one promising solution, as they visually represent logistic regression results in a concise, intuitive, and quantitative format that can be readily applied in clinical settings (Kong et al., 2019; Pan et al., 2019; Balachandran et al., 2015).

In this study, we collected clinical data, laboratory test results, and three-month follow-up information for AIS patients who received intravenous rt-PA therapy. We then constructed a risk prediction model that may assist clinicians in making informed decisions both before and after thrombolysis.

Materials & Methods

Research design and participant selection

This retrospective study enrolled patients with AIS who underwent intravenous rt-PA therapy at the Third Central Hospital of Tianjin between December 2016 and December 2022. All patients signed the informed consent. The protocol was approved by the Medical Ethics Committee of the Third Central Hospital of Tianjin. No formal ethics number was issued, and all participants provided informed consent. Follow-up evaluations were conducted in outpatient clinics 1, 3, and 6 months after discharge.

Exclusion criteria

Patients were excluded if imaging studies (CT or MRI) revealed evidence of hemorrhage, multilobar infarction, neoplasms, or vascular malformations. Individuals who had a history of severe extracranial trauma or intracranial surgery were also excluded. We further excluded those with coagulation abnormalities who had been receiving routine anticoagulant therapy prior to enrollment, as well as patients with severe cardiac, hepatic, or renal conditions, malignant tumors, epileptic seizures, or acute psychiatric episodes. Individuals who did not complete follow-up or who underwent additional endovascular procedures during treatment were likewise excluded.

Data collection

Clinical variables were extracted from the electronic medical record system, including demographic and anthropometric information (age, sex, height, weight), blood pressure (systolic and diastolic) measured at the time of thrombolysis, tobacco use, medical history (hypertension, coronary artery disease, diabetes, atrial fibrillation (AF), hyperlipidemia (HL), previous stroke), time from symptom onset to intravenous rt-PA administration, infarct location, and complications such as intracranial hemorrhage following thrombolysis. We also recorded baseline NIHSS scores, three-month modified Rankin Scale (mRS) scores, and laboratory data at admission, including complete blood counts (white blood cells, red blood cells, hemoglobin, platelets, neutrophil percentage), electrolytes (potassium, sodium), coagulation parameters (PT, APTT, INR, Fg), and renal function measures (urea nitrogen (UN), creatinine).

Outcome assessment

Stroke severity was evaluated using the National Institutes of Health Stroke Scale (Brott et al., 1989), and prognosis was determined three months after treatment by applying the mRS (Bamford et al., 1989). The mRS classifies a patient’s condition according to the following criteria: a score of 0 indicates no symptoms; a score of 1 denotes minor symptoms without functional limitations, allowing individuals to manage everyday life independently; a score of 2 corresponds to mild disability, limiting some previously routine tasks but still permitting most daily activities; a score of 3 represents moderate disability, requiring help with daily activities other than walking; a score of 4 signifies severe disability, necessitating assistance with nearly all daily tasks; a score of 5 describes a bedridden state with incontinence and a need for ongoing care; and a score of 6 indicates death. An mRS score of 2 or lower was considered a favorable prognosis, whereas a score above 2 was defined as unfavorable (Sulter, Steen & De Keyser, 1999).

Sample size estimation

Sample size was determined using the events per variable (EPV) method, requiring at least 10 positive outcomes per predictor variable in the training set. With five anticipated predictor variables, a minimum of 50 cases with poor outcomes was necessary. Given an expected poor outcome rate of 20% at three months, the training set required at least 250 cases. Based on a predetermined 7:3 allocation ratio between training and validation sets, the total sample size was calculated at 357 cases. A 10% adjustment for potential loss to follow-up yielded a final target enrollment of 392 patients.

Missing data management

The dataset contained minimal missing values, which were handled through multiple imputation.

Statistical methods

Study participants were randomly allocated to training and validation sets in a 7:3 ratio. The development of the predictive model involved identifying independent predictors through univariate and stepwise multivariate logistic regression analyses in the training set, followed by nomogram construction. Model performance was evaluated using the area under the receiver operating characteristic (ROC) curve, while calibration was assessed through plots comparing predicted and observed outcomes. Decision curve analyses (DCA) and clinical impact curves provided insights into the model’s clinical utility.

Continuous variables are presented as mean ± standard deviation, with between-group comparisons conducted using independent-sample t-tests. Categorical variables are expressed as frequencies and percentages, with group comparisons performed using Chi-square tests or Fisher’s exact tests as appropriate. Statistical significance was set at a two-sided P value < 0.05. All analyses were conducted using R version 4.3.1.

Results

Fundamental characteristics of the training and validation sets

A total of 392 participants were enrolled, with 275 assigned to the training set and 117 to the validation set. Among all participants, 63% were male and 37% were female, and the mean age was 67.94 ± 11.01 years. The average baseline NIHSS score was 6.96 ± 5.43. Overall, 77 participants (60 in the training set and 17 in the validation set) had poor outcomes (Table 1).

Table 1 Basic demographic information and laboratory marker characteristics in the training and validation sets.

No	Factors	Training (n = 275)	Validation (n = 117)	Overall (n = 392)	
1	Gender				
Male	159 (57.8)	88 (75.2)	247 (63.0)	
Female	116 (42.2)	29 (24.8)	145 (37.0)	
2	Ages	71.65 ± 8.41	59.23 ± 11.5	67.94 ± 11.01	
5	BMI	24.51 ± 2.95	25.4 ± 3.43	24.77 ± 3.13	
6	Onset Time	159.15 ± 56.61	158.55 ± 53.77	158.97 ± 55.71	
7	NIHSS	7.17 ± 5.72	6.48 ± 4.65	6.96 ± 5.43	
8	Smoking				
No	142 (51.6)	42 (35.9)	184 (46.9)	
Yes	133 (48.4)	75 (64.1)	208 (53.1)	
9	Hypertension				
No	78 (28.4)	40 (34.2)	118 (30.1)	
Yes	197 (71.6)	77 (65.8)	274 (69.9)	
10	Prior Atrial Fibrillation				
No	221 (80.4)	105 (89.7)	326 (83.2)	
Yes	54 (19.6)	12 (10.3)	66 (16.8)	
11	Prior Ischemic Heart Disease				
No	244 (88.7)	111 (94.9)	355 (90.6)	
Yes	31 (11.3)	6 (5.1)	37 (9.4)	
12	Incident Atrial Fibrillation				
No	268 (97.5)	111 (94.9)	379 (96.7)	
Yes	7 (2.5)	6 (5.1)	13 (3.3)	
13	Diabetes Mellitus				
No	190 (69.1)	74 (63.2)	264 (67.3)	
Yes	85 (30.9)	43 (36.8)	128 (32.7)	
14	Hyperlipidemia				
No	195 (70.9)	71 (60.7)	266 (67.9)	
Yes	80 (29.1)	46 (39.3)	126 (32.1)	
15	Coronary Heart Disease				
No	195 (70.9)	102 (87.2)	297 (75.8)	
Yes	80 (29.1)	15 (12.8)	95 (24.2)	
16	Congestive Heart Failure				
No	263 (95.6)	116 (99.1)	379 (96.7)	
Yes	12 (4.4)	1 (0.9)	13 (3.3)	
17	Stroke History				
No	212 (77.1)	107 (91.5)	319 (81.4)	
Yes	63 (22.9)	10 (8.5)	73 (18.6)	
18	CHD History				
No	274 (99.6)	117 (100)	391 (99.7)	
Yes	1 (0.4)	0 (0)	1 (0.3)	
19	SBP	151.2 ± 14.88	152.09 ± 14.04	151.46 ± 14.62	
20	DBP	82.77 ± 10.31	87.12 ± 10.22	84.07 ± 10.46	
21	Hb	135.37 ± 15.65	141.49 ± 15.97	137.19 ± 15.97	
22	RBC	4.41 ± 0.52	4.6 ± 0.52	4.47 ± 0.52	
23	WBC	6.95 ± 2.31	7.09 ± 2.5	6.99 ± 2.37	
24	N	66.5 ± 10.45	64.69 ± 10.39	65.96 ± 10.45	
25	PLT	208.52 ± 59.07	208.79 ± 61.71	208.6 ± 59.79	
26	K	3.9 ± 0.38	3.92 ± 0.36	3.91 ± 0.37	
29	Na	141.57 ± 3.2	141.89 ± 2.73	141.67 ± 3.06	
30	UN	5.53 ± 1.83	5.14 ± 1.32	5.42 ± 1.7	
31	Cr	75.54 ± 23.55	72.72 ± 17	74.7 ± 21.82	
32	PT	13.01 ± 0.82	12.67 ± 0.76	12.91 ± 0.81	
33	APTT	35.27 ± 3.43	34.51 ± 3.34	35.04 ± 3.41	
34	INR	1 ± 0.08	0.98 ± 0.07	0.99 ± 0.08	
35	Fg	3.08 ± 0.97	2.92 ± 0.73	3.04 ± 0.91	
36	Complications				
No	255 (92.7)	113 (96.6)	368 (93.9)	
Yes	20 (7.3)	4 (3.4)	24 (6.1)	
37	InfarctionSite				
1	195 (70.9)	81 (69.2)	276 (70.4)	
2	74 (26.9)	36 (30.8)	110 (28.1)	
3	6 (2.2)	0 (0)	6 (1.5)	
38	Group				
0	215 (78.2)	100 (85.5)	315 (80.4)	
1	60 (21.8)	17 (14.5)	77 (19.6)	

Variable selection

Univariate analyses revealed significant differences (P < 0.05) between the good and poor outcome groups in factors such as age, baseline NIHSS, a history of AF, HL, chronic heart failure, pre-thrombolysis systolic blood pressure, leukocyte count at admission, neutrophil percentage, serum sodium, UN, fibrinogen (Fg), occurrence of intracranial hemorrhage following thrombolysis, and infarct location. Moreover, the history of hypertension and family history of coronary heart disease approached statistical significance, suggesting their potential influence on adverse prognosis could not be discounted (Table 2). Accordingly, all these variables were incorporated into the stepwise multivariate logistic regression analysis.

Table 2 Distribution differences of variables between good and poor prognosis groups in the training set.

No	Factors	Group 0	Group 1	Overall	Statistic	P	
1	Gender				1.923	0.165	
Male	129 (60)	30 (50)	159 (57.8)			
Female	86 (40)	30 (50)	116 (42.2)			
2	Ages	70.24 ± 7.93	76.7 ± 8.22	71.65 ± 8.41	−5.53	<0.001	
5	BMI	24.57 ± 2.9	24.28 ± 3.16	24.51 ± 2.95	0.668	0.504	
6	Onset Time	161.44 ± 55.35	150.92 ± 60.68	159.15 ± 56.61	1.275	0.203	
7	NIHSS	4.96 ± 3.19	15.08 ± 5.81	7.17 ± 5.72	−17.745	<0.001	
8	Smoking				2.15	0.143	
No	106 (49.3)	36 (60)	142 (51.6)			
Yes	109 (50.7)	24 (40)	133 (48.4)			
9	Hypertension				3.8	0.051	
No	67 (31.2)	11 (18.3)	78 (28.4)			
Yes	148 (68.8)	49 (81.7)	197 (71.6)			
10	Prior Atrial Fibrillation				14.105	<0.001	
No	183 (85.1)	38 (63.3)	221 (80.4)			
Yes	32 (14.9)	22 (36.7)	54 (19.6)			
11	Prior Ischemic Heart Disease				1.066	0.302	
No	193 (89.8)	51 (85)	244 (88.7)			
Yes	22 (10.2)	9 (15)	31 (11.3)			
12	Incident Atrial Fibrillation				1.864	0.172	
No	211 (98.1)	57 (95)	268 (97.5)			
Yes	4 (1.9)	3 (5)	7 (2.5)			
13	Diabetes Mellitus				1.255	0.263	
No	145 (67.4)	45 (75)	190 (69.1)			
Yes	70 (32.6)	15 (25)	85 (30.9)			
14	Hyperlipidemia				4.428	0.035	
No	159 (74)	36 (60)	195 (70.9)			
Yes	56 (26)	24 (40)	80 (29.1)			
15	Coronary Heart Disease				2.135	0.144	
No	157 (73)	38 (63.3)	195 (70.9)			
Yes	58 (27)	22 (36.7)	80 (29.1)			
16	Congestive Heart Failure				14.795	<0.001	
No	211 (98.1)	52 (86.7)	263 (95.6)			
Yes	4 (1.9)	8 (13.3)	12 (4.4)			
17	Stroke History				0.91	0.34	
No	163 (75.8)	49 (81.7)	212 (77.1)			
Yes	52 (24.2)	11 (18.3)	63 (22.9)			
18	CHD History				3.596	0.058	
No	215 (100)	59 (98.3)	274 (99.6)			
Yes	0 (0)	1 (1.7)	1 (0.4)			
19	SBP	150.16 ± 14.8	154.9 ± 14.72	151.2 ± 14.88	−2.195	0.029	
20	DBP	82.41 ± 9.95	84.07 ± 11.52	82.77 ± 10.31	−1.101	0.272	
21	Hb	135.38 ± 15.55	135.33 ± 16.16	135.37 ± 15.65	0.019	0.985	
22	RBC	4.4 ± 0.51	4.44 ± 0.56	4.41 ± 0.52	−0.498	0.619	
23	WBC	6.67 ± 2.12	7.97 ± 2.69	6.95 ± 2.31	−3.957	<0.001	
24	N	64.48 ± 9.36	73.74 ± 11.02	66.5 ± 10.45	−6.515	<0.001	
25	PLT	207.51 ± 57.59	212.13 ± 64.48	208.52 ± 59.07	−0.535	0.593	
26	K	3.92 ± 0.38	3.82 ± 0.39	3.9 ± 0.38	1.811	0.071	
27	Na	141.83 ± 2.92	140.64 ± 3.92	141.57 ± 3.2	2.593	0.01	
28	UN	5.41 ± 1.68	5.99 ± 2.24	5.53 ± 1.83	−2.193	0.029	
29	Cr	75.7 ± 23.29	74.95 ± 24.67	75.54 ± 23.55	0.218	0.827	
30	PT	12.97 ± 0.82	13.15 ± 0.78	13.01 ± 0.82	−1.528	0.128	
31	APTT	35.41 ± 3.5	34.79 ± 3.13	35.27 ± 3.43	1.236	0.218	
32	INR	0.99 ± 0.08	1.01 ± 0.08	1 ± 0.08	−1.127	0.261	
33	Fg	2.96 ± 0.77	3.54 ± 1.39	3.08 ± 0.97	−4.284	<0.001	
34	Complications				4.18	0.041	
No	203 (94.4)	52 (86.7)	255 (92.7)			
Yes	12 (5.6)	8 (13.3)	20 (7.3)			
35	InfarctionSite				11.308	0.004	
1	143 (66.5)	52 (86.7)	195 (70.9)			
2	68 (31.6)	6 (10)	74 (26.9)			
3	4 (1.9)	2 (3.3)	6 (2.2)			

Nomogram construction

Stepwise logistic regression ultimately identified three independent predictors of poor outcomes following intravenous alteplase therapy in patients with AIS: baseline NIHSS, Fg levels, and a history of hypertension. A nomogram-based risk prediction model for adverse prognosis was developed for AIS patients after alteplase intravenous thrombolytic therapy using these three variables (Fig. 1).

Figure 1 Nomogram for predicting adverse prognosis 3 months after intravenous rt-PA therapy.

Nomogram validation

In the training set, the area under the ROC curve was 0.948 (95% CI [0.910–0.985]), with a sensitivity of 0.900 and a specificity of 0.916 at a high-risk probability threshold of 0.219 (Fig. 2). Calibration curves demonstrated a substantial degree of calibration, with pre-and-post calibration curves closely adhering to the optimal curve (Fig. 3). DCA showed a net benefit over threshold probabilities ranging from 0.18 to 0.85 (Fig. 4). In addition, the clinical impact curve based on 1,000 hypothetical samples confirmed that the model’s predicted high-risk group closely matched the actual high-risk group at a threshold probability of 0.2 (Fig. 5).

Figure 2 ROC curve of the training set.

Figure 3 Calibration curve of the training set.

Figure 4 Clinical decision curve of the training set.

Figure 5 Clinical impact curve of the training set.

Model validation

In the validation set, the area under the ROC curve was 0.959 (95% CI [0.907–1.000]), with a sensitivity of 0.943 and a specificity of 0.947 (Fig. 6). Calibration plots again showed close agreement between observed and predicted probabilities (Fig. 7). DCA indicated a wide net benefit range from threshold probabilities of 0.18 to 0.9 (Fig. 8). Finally, the clinical impact curve, derived from 1,000 hypothetical samples, demonstrated that the nomogram’s predicted high-risk group accurately reflected the actual high-risk group at a threshold probability of 0.2 (Fig. 9).

Figure 6 ROC curve of the validation set.

Figure 7 Calibration curve of the validation set.

Figure 8 Clinical decision curve of the validation set.

Figure 9 Clinical impact curve of the validation set.

Discussion

The study developed and validated a prognostic nomogram based on demographic and clinical indicators to predict three-month adverse outcomes in patients with AIS who received intravenous rt-PA. Although disease severity is the most influential factor in determining a patient’s prognosis, accurately identifying elements associated with poor outcomes remains critical for guiding a range of clinical decisions.

A key strength of this nomogram lies in its ability to provide individualized estimates of a patient’s likelihood of experiencing a poor outcome. From a clinical perspective, this approach offers an easily interpretable, quantitative tool for discussing prognosis with patients and their families. The clinical utility of the nomogram was evaluated through DCA, an established method for assessing predictive models (Rousson & Zumbrunn, 2011; Vickers & Elkin, 2006). The nomogram successfully differentiated between high-risk and low-risk patients, with net benefit analysis demonstrating its superiority over NIHSS score alone across various probability thresholds. These findings support the clinical value of the nomogram in outcome prediction and early identification of patients who may benefit from additional therapeutic interventions.

To mitigate the issues of overfitting and biased distributions often encountered with conventional logistic regression, demographic characteristics, vital signs, complications, and laboratory indicators were incorporated into the regression model. Although machine learning methods have received extensive attention for disease diagnosis, prognosis, and treatment response prediction, they must balance interpretability and accuracy. Models with better interpretability, such as logistic regression, ordinal regression, and Cooks regression, often achieve lower accuracy, while less interpretable models (for example, neural networks or random forests) generally show higher accuracy. In clinical settings, however, scoring tools typically prioritize interpretability. In our investigation, a nomogram derived solely from logistic regression achieved excellent predictive performance while maintaining excellent interpretability, clearly illustrating how each factor contributes to the risk of a poor outcome. This dual advantage of accuracy and interpretability provide valuable insights for the future development of standardized scoring tools.

The nomogram incorporating baseline NIHSS, Fg, and a history of hypertension demonstrated excellent discrimination, calibration, and clinical utility. The predictor variables in this model are routinely measured in clinical practice and are readily available. Most of these variables, such as age, sex, NIHSS, AF, and history of stroke, have also been reported in previous studies as risk factors for adverse outcomes in AIS (Roy-O’Reilly & McCullough, 2018; Kwah & Diong, 2014; Keller et al., 2020; Patti et al., 2019).

NIHSS is a well-established measure for evaluating neurological deficits and predicting outcomes in stroke patients. Higher NIHSS scores typically indicate a poorer prognosis (Kwah & Diong, 2014; Chalos et al., 2020). Because NIHSS scoring at admission reflects initial stroke severity, it often appears in predictive models for AIS prognosis. Early clinical responses after thrombolysis can account for a large proportion of long-term outcomes. Consistent with previous research, the present model identifies NIHSS as its most influential predictor.

Beyond stroke severity, age also appears to be a key factor in AIS outcomes (Roy-O’Reilly & McCullough, 2018). While earlier clinical trials found that older patients fared worse than younger patients after rt-PA thrombolysis, more recent evidence has been inconsistent. In a pooled analysis of seven randomized trials, Emberson et al. (2014) suggested that advanced age does not necessarily reduce the likelihood of a favorable outcome post-rt-PA (Roy-O’Reilly & McCullough, 2018). Conflicting findings may be explained by differences in how outcomes are measured. Moreover, older adults are more likely to have comorbidities such as AF and hypertension, along with higher NIHSS scores, potentially affecting stroke outcomes. Although patients with unfavorable outcomes in our study were generally older, the age effect did not remain significant once we accounted for potential confounding variables.

Several studies have explored the relationship between sex and AIS outcomes (Roy-O’Reilly & McCullough, 2018; Rodríguez-Castro et al., 2019; Thilarajah et al., 2018; Purroy et al., 2021). In women, stroke risk increases substantially after menopause, typically after age 42 (Lisabeth et al., 2009). Women over 80 are at higher risk of ischemic stroke, with an odds ratio of 2.27 (95% CI [1.71–3.02]) (Rojas et al., 2007), possibly owing to the protective effect of estrogen (Hurn & Macrae, 2000). Nonetheless, our study, in line with a Chinese cohort study of AIS patients treated with intravenous thrombolysis (Zhou et al., 2021), did not identify sex as an independent risk factor for poor prognosis. Additional subgroup analyses are warranted to confirm this finding, especially since potential racial differences may also play a role.

AF can form cardioembolic clots that obstruct intracranial blood vessels, initiating AIS. Numerous studies recognize AF as a risk factor for AIS (Keller et al., 2020; Hsu, Huang & Lin, 2020). Interestingly, our results indicate that AF did not significantly influence prognosis following thrombolytic therapy. This discrepancy may be linked to the subtype of cerebral infarction, specifically whether AF was the direct cause of stroke. Further investigations with larger sample sizes and longer follow-up periods are needed to validate this hypothesis.

Patients who have had a previous stroke often exhibit intracranial vascular stenosis, compromising collateral circulation and affecting overall stroke prognosis. Several studies have reported that a history of stroke substantially influences the outcome of AIS (Leker et al., 2019; Ogawa et al., 2017). In our nomogram, however, a previous stroke did not emerge as a significant factor in predicting poor outcomes after rt-PA thrombolysis. One reason may be that earlier strokes in our cohort were relatively mild and did not lead to severe disability. Still, it remains plausible that multiple prior strokes could worsen long-term outcomes, a possibility that should be explored in future investigations.

Although hypertension is widely recognized as a risk factor for stroke, Potter et al. (2009) observed that stroke patients whose blood pressure was intensively controlled post-stroke faced higher 90-day mortality than those who did not receive antihypertensive treatment. This phenomenon may stem from diminished autoregulatory capacity of major arteries following stroke and the need for higher systolic pressure to maintain perfusion in the ischemic penumbra. Indeed, studies have shown that patients with decreased vessel elasticity often have significant arterial narrowing, whereas mild or moderate hypertension may promote increased blood flow to the penumbra and, consequently, improve outcomes (Faraci & Heistad, 1990). Our findings similarly indicate that a history of hypertension is associated with better outcomes after intravenous alteplase therapy.

The severity and location of infarction are primary determinants of AIS prognosis. The NIHSS is an internationally recognized scale for quantifying the severity of acute ischemic stroke (Cincura et al., 2009). However, the NIHSS has notable limitations—it assigns higher scores to left hemisphere infarctions compared to right hemisphere infarctions, and to anterior circulation compared to posterior circulation infarctions (Sato et al., 2008; Kazi, Siddiqui & Majid, 2021). This scoring pattern can create a misleading impression that posterior circulation infarctions yield better functional outcomes. Previous research has revealed distinct patterns in the relationship between NIHSS scores and functional prognosis in anterior versus posterior circulation infarctions. In posterior circulation infarctions, even low NIHSS scores may be associated with poor functional outcomes. Conversely, in anterior circulation infarctions, patients with favorable outcomes often show higher mean NIHSS scores than their posterior circulation counterparts. Notably, the mean NIHSS scores for poor functional outcomes in posterior circulation infarctions approximate those associated with favorable outcomes in anterior circulation infarctions (Kazi, Siddiqui & Majid, 2021). Therefore, prognosis should not be assessed solely based on circulation territory. This study confirms that the location of infarction does not serve as a significant predictor of poor outcomes in patients receiving rt-PA thrombolysis, consistent with previous findings.

Reperfusion therapies such as intravenous thrombolysis and endovascular thrombectomy markedly improve prognosis for patients with AIS (Hacke et al., 2008; Powers et al., 2018; Zi et al., 2021). However, intravenous thrombolysis elevates the risk of hemorrhagic transformation, which can independently worsen functional recovery and increase mortality (Von Kummer et al., 2015). Previous studies report that 10–48% of rt-PA–treated AIS patients may develop hemorrhagic transformation, and 2–7% may experience symptomatic intracranial hemorrhage (Emberson et al., 2014; Lindley et al., 2004). In our study, intracranial hemorrhage initially appeared to correlate significantly with unfavorable outcomes, yet further analysis did not confirm a robust link between these events and poor prognosis. This discrepancy may be attributable to patient selection biases inherent in single-center designs and underscores the need for larger multicenter trials to validate our findings.

Fg, a key protein influencing coagulation function and a major determinant of blood viscosity, contributes to platelet aggregation, primary hemostasis, and interactions between leukocytes and endothelial cells (Tanne et al., 2006). Elevated Fg levels in patients with AIS are linked to worse neurological outcomes and diminished thrombolytic efficacy, likely because high Fg levels increase thrombotic resistance to fibrinolysis and raise the risk of re-occluding the affected vessel (Lu et al., 2018). A study by Lu et al. (2018) found that early reductions in Fg levels after rt-PA treatment were independently associated with better short-term outcomes, with the odds of functional improvement increasing by a factor of 3.84 for every 20% decrease in Fg levels. Consistent with previous reports, our findings indicate that higher Fg levels are associated with poorer outcomes at three months post-thrombolysis. Hence, Fg may serve as a useful prognostic biomarker for guiding short-term treatment goals and therapeutic strategies after rt-PA administration (Shi et al., 2020).

Growing evidence also suggests that certain hematological parameters, including total white blood cell count, neutrophil count, lymphocyte count, and the neutrophil-to-lymphocyte ratio, may play pivotal roles in stroke progression (Maestrini et al., 2015). Observational studies in patients receiving rt-PA have shown that lower white blood cell counts at admission correlate with early neurological recovery and better functional outcomes at 90 days (Malhotra et al., 2018; Chen et al., 2020). However, few studies have examined whether these baseline values also affect the success of thrombolytic therapy. A single-center retrospective analysis reported that higher white blood cell and neutrophil counts were associated with an elevated risk of hemorrhagic transformation and poorer outcomes at three months following rt-PA treatment, although it found no clear relationship between admission white blood cell count and three-month clinical status (Xie et al., 2023). In this study, patients with unfavorable outcomes had higher white blood cell and neutrophil counts than those with favorable outcomes. Yet, further statistical analyses indicated that neither white blood cell count nor neutrophil count was an independent predictor of poor prognosis after rt-PA thrombolysis. These results align with the aforementioned single-center retrospective study, underscoring the complexity of hematological contributions to stroke recovery.

This study has several limitations. For instance, the hospitalization of patients may depend on factors such as the clinical judgment of the attending physician and the financial resources of each patient, which might introduce selection bias. In addition, this was a single-center, retrospective cohort study that focused on patients with a mean age of 67.94 ± 11.01 years who received only rt-PA thrombolysis, so the sample size was relatively small. Although our nomogram underwent extensive internal validation, including bootstrapping, and exhibited near-ideal performance in both training and validation sets, the possibility of overfitting cannot be completely excluded. Future multicenter studies with larger cohorts will be necessary to confirm these findings. The variables used in our model were chosen based on their routine availability in clinical practice, ensuring that the nomogram is convenient and easy to implement. Nevertheless, adverse disease outcomes may also be influenced by lifestyle factors and social determinants that are not readily captured in real-world settings, potentially reducing the accuracy of the model. The absence of an independent external validation cohort further limits the generalizability of the results. Future research will incorporate data from multiple centers to enhance and validate the model. Another aspect left unexplored in this study is the influence of alternative treatment strategies, such as endovascular intervention, on patient prognosis. Although the nomogram offers a rapid way to identify individuals at higher risk of poor three-month outcomes after AIS, it does not explain the underlying pathophysiological mechanisms. Despite these limitations, this study presents a simple, effective, and personalized nomogram to predict the risk of unfavorable outcomes at 3 months in patients with AIS receiving rt-PA thrombolysis.

Conclusion

In this study, we created a nomogram to predict the likelihood of a poor three-month prognosis in patients with AIS who undergo rt-PA thrombolysis. Developed using easily collected demographic and clinical data, the nomogram exhibits strong discrimination, acceptable calibration, and a wide range of net benefit thresholds. Consisting of only three key variables, it can be applied readily at the bedside. This tool may serve as a valuable addition to current assessments of prognosis in AIS patients treated with alteplase, providing an evidence-based foundation for more effective clinical decision-making and earlier rehabilitation planning.

Supplemental Information

Data S1 Raw data

Additional Information and Declarations

Competing Interests

Author Contributions

Human Ethics

Ethics

Data Availability

The authors declare there are no competing interests.

Fengjiao Zhang conceived and designed the experiments, performed the experiments, analyzed the data, prepared figures and/or tables, authored or reviewed drafts of the article, and approved the final draft.

Dan Zhao conceived and designed the experiments, performed the experiments, analyzed the data, prepared figures and/or tables, authored or reviewed drafts of the article, and approved the final draft.

Jing Zhang conceived and designed the experiments, authored or reviewed drafts of the article, and approved the final draft.

The following information was supplied relating to ethical approvals (i.e., approving body and any reference numbers):

Third Central Hospital of Tianjin.

The following information was supplied relating to ethical approvals (i.e., approving body and any reference numbers):

Medical Ethics Committee of the Third Central Hospital of Tianjin.

The following information was supplied regarding data availability:

The raw data is available in the Supplemental File.

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
