# Peer review of "Development and validation of a prognostic nomogram for predicting poor outcomes following intravenous rt-PA in patients with acute ischemic stroke"

_PeerJ, doi:10.7717/peerj.18937_

## Round 0.1 · original submission · Major Revisions

Dear Dr. Zhang,


Your manuscript has been revised by two experts in the field and both have identified several critical areas requiring improvement. While the manuscript demonstrates scientific rigor and addresses a clinically relevant question, the following major issues must be addressed to improve its quality and impact:

1. Basic Reporting
Language and Grammar: Simplify overly complex phrasing and correct grammatical errors (e.g., "unfavorable outcome" should be "grim prognosis"). Maintain consistent terminology throughout.
Figures and Tables: Improve the resolution of Figures 1 and 4. Provide more descriptive captions and better explanations for standalone comprehension. Summarize or reformat dense tables (e.g., Table 1) for clarity.
Background and Justification: Justify the choice of NIHSS, fibrinogen, and hypertension as model predictors. Compare the proposed model with existing prognostic tools and highlight its novelty.
Discussion and Novelty: Discuss how the nomogram advances existing models and explicitly identify its unique contributions.

2. Experimental Design
Study Design and Validation: The retrospective nature of the study introduces inherent biases. Address how these biases are mitigated and outline a plan for future external validation in multi-center cohorts.
Statistical Methods: Justify the use of stepwise logistic regression, as it is often criticized for its susceptibility to overfitting. Report the sample size calculation and provide justification for its adequacy.
Predictor Selection and Missing Data: Clarify why key predictors (e.g., age, sex, glucose levels) were excluded. Provide transparency on the handling of missing data (e.g., imputation methods) and assess its impact on the results.

3. Validity of the Findings
Claims of Clinical Utility: Avoid overstating the clinical relevance of the findings, particularly due to the absence of external validation. Adjust claims of the nomogram’s "superiority" to reflect the limitations of a single-center study.
Model Generalizability: Address the potential for overfitting and discuss plans to validate the nomogram on larger, more diverse populations. Acknowledge the impact of demographic differences on model performance.
Clinical Integration: Discuss how the nomogram can be integrated into clinical workflows. Specify threshold values for decision-making and provide clinical scenarios illustrating its potential impact on patient care.
Actionable Recommendations
Language and Presentation: Revise the manuscript for grammar, clarity, and consistency. Update figure and table captions to be more descriptive. Improve figure resolution.
Background and Rationale: Strengthen the justification for the choice of predictors. Provide a comparison with existing models and clearly state how the nomogram advances the field.
Methodological Clarity: Address gaps in the study design (e.g., external validation, rationale for predictor exclusion, missing data handling, and stepwise logistic regression use).
Claims of Novelty and Utility: Tone down claims of novelty and utility. Discuss generalizability and potential demographic impacts.
Clinical Relevance and Integration: Provide specific examples of clinical integration and decision-making thresholds. Discuss next steps for multi-center validation.

Thus, I ask you to address these major issues which will significantly enhance the robustness, clarity, and clinical relevance of the manuscript.

·

Basic reporting

- The manuscript uses clear language but includes minor grammatical errors. Phrases like "grim prognosis" should replace "unfavorable outcome."
- Figures 1 and 4 require better resolution.
- Tables need more descriptive captions.
- Background is adequate, but the justification for focusing on NIHSS, fibrinogen, and hypertension is weak. Comparison with other models is lacking.

Experimental design

- Retrospective design is not ideal for developing a prognostic model due to inherent biases and lack of prospective validation. The absence of external validation further undermines generalizability.
-Stepwise logistic regression is used without justification. Exclusion criteria lack clarity.

Validity of the findings

- Claims about clinical utility are exaggerated given the lack of prospective or external validation. Findings may not translate well to real-world applications.

Additional comments

None

Reviewer 2 ·

Basic reporting

The manuscript is generally well-written and uses professional language. However, there are instances of overly complex phrasing and grammatical issues (e.g., "the prognosis was ascertained via the mRS score at a 3-month post-treatment mark"). Simplifying sentences and ensuring consistent terminology would improve readability.

The introduction provides a good overview of the context, highlighting the importance of predicting outcomes after rt-PA treatment. However, the background could be expanded to discuss other existing prognostic models and their limitations, which would better justify the study’s novelty.

The manuscript follows a standard structure with clearly labeled sections for introduction, methods, results, and discussion. The figures and tables are well-organized and relevant to the study. However, some tables (e.g., Table 1) contain dense information that could benefit from summarization or clearer formatting and figure legends could include more detailed explanations for better standalone comprehension The raw data is provided, adhering to PeerJ's data-sharing policy.

The manuscript is self-contained and addresses the stated hypothesis effectively. The inclusion of both training and validation datasets strengthens the study's reliability. However, the lack of external validation remains a limitation that should be explicitly acknowledged.

Questions:
1. While the nomogram demonstrates high discrimination (AUC 0.948 for the training set, 0.937 for validation), external validation is missing. The authors should discuss how they plan to validate this model in multi-center cohorts to ensure generalizability.
2. Although appropriate statistical methods were used, the manuscript lacks a power calculation or justification for the sample size. This omission weakens the robustness of the findings.
3. How does your nomogram improve upon existing predictive models for stroke prognosis? Could you provide a direct comparison of predictive accuracy or usability?
4. The authors highlight the practical use of the nomogram but fail to discuss its integration into clinical workflows. What are the proposed threshold values for clinical use, and how do these compare to existing standards? Could you provide specific scenarios where this nomogram would change clinical decision-making?
5. While the study contributes to the field of stroke prognosis, its novelty is limited because similar nomograms have been developed. The authors need to present the specific advancements or improvements their model offers over existing tools.
6. Have you considered validating the nomogram in a larger, more diverse population? How do you anticipate differences in demographics or stroke care protocols to affect the model's performance?

Experimental design

The study is within aims and scope of the journal.

The research question is clearly stated and the study addresses a clinically relevant knowledge gap.

The investigation appears rigorous with a well-documented methodology. However, the lack of external validation is a notable limitation that the authors should address.

The methods are described in sufficient detail, allowing replication by other researchers. However, additional clarity on certain aspects, such as the handling of missing data and the rationale for specific variable exclusions, would improve transparency.

Questions:
1. Could you elaborate on why some commonly used predictors in stroke prognosis (e.g., age, sex, glucose levels) were excluded from the final model?
2. How were missing data handled in the study? If imputation methods were used, what was their impact on the results?

Validity of the findings

While this study is not explicitly a replication, its focus on nomogram development aligns with ongoing efforts in the field. The study's contribution lies in refining and validating a prediction model using routine clinical variables, which is useful but not groundbreaking.

Raw data and validation cohorts are described, supporting transparency. However, details about how missing data were managed (if any) are lacking, and this could influence replicability and interpretation.

Statistical analyses are appropriate and robust. However, external validation is missing, which limits the generalizability of the findings.

The conclusions align with the results, focusing on the nomogram’s utility for predicting poor outcomes. However, some statements (e.g., the model's superiority) should be tempered given the single-center nature of the study.

The authors should discuss the potential impact of their nomogram in clinical decision-making in more detail.

Limitations are acknowledged but could be expanded, particularly regarding the lack of external validation and the potential for overfitting in the model.

---

## Round 0.2 · accepted · Accept

Dear Dr. Zhang,

Congratulations on the acceptance of your manuscript.

·

Basic reporting

Thank you for addressing my concern

Experimental design

Thank you for addressing my concern

Validity of the findings

Thank you for addressing my concern

Additional comments

Thank you for addressing my concern

Reviewer 2 ·

Basic reporting

I think authors addressed my questions and concerns. I do not have more questions.

Experimental design

I think authors addressed my questions and concerns. I do not have more questions.

Validity of the findings

I think authors addressed my questions and concerns. I do not have more questions.